# Interaction Order Estimation in Tensor Curie–Weiss Models

**DOI:** 10.3390/e27030245

**Published:** 2025-02-27

**Authors:** Somabha Mukherjee

**Affiliations:** Department of Statistics and Data Science, Faculty of Science, National University of Singapore, Singapore 117546, Singapore; somabha@nus.edu.sg

**Keywords:** Curie–Weiss model, joint estimation, interaction order, contiguity

## Abstract

In this paper, we consider the problem of estimating the interaction parameter *p* of a *p*-spin Curie–Weiss model at inverse temperature β, given a single observation from this model. We show, by a contiguity argument, that joint estimation of the parameters β and *p* is impossible, which implies that the estimation of *p* is impossible if β is unknown. These impossibility results are also extended to the more general *p*-spin Erdös–Rényi Ising model. The situation is more delicate when β is known. In this case, we show that there exists an increasing threshold function $\beta^{*}\left(p\right)$, such that for all β, consistent estimation of *p* is impossible when $\beta^{*}\left(p\right)>\beta$, and for *almost all*
β, consistent estimation of *p* is possible for $\beta^{*}\left(p\right)<\beta$.

## 1. Introduction

The Ising model [1] was originally introduced in Physics as a model for ferromagnetism, and has since then found numerous interesting applications in diverse areas, such as image processing [2], neural networks [3], spatial statistics [4], and disease mapping in epidemiology [5]. The classical (2-spin) Ising model is a discrete probability measure on the set of all binary strings of a fixed length, given by the following:(1)$$P_{\beta }\left(x\right):=\frac{\exp\left\{-\beta H_{N}\left(x\right)\right\}}{Z_{N}\left(\beta \right)}\;\mathrm{for}\;x\in {\left\{-1,1\right\}}^{N}$$
where the *Hamiltonian* $H_{N}\left(x\right)$ is given by$$H_{N}\left(x\right):=-\sum_{1\leq i,j\leq N}J_{i,j}x_{i}x_{j}\;,$$
$J:={\left(\left(J_{i,j}\right)\right)}_{1\leq i,j\leq N}$ is the interaction matrix (often taken as the scaled adjacency matrix of some graph), β>0 denotes the interaction strength (called *inverse temperature* in the physics literature), and $Z_{N}\left(\beta \right)$ is the normalizing constant, needed to ensure that the probabilities in (Equation 1) add up to 1 (referred to as the *partition function* in Physics). The *Hamiltonian*$\sum_{1\leq i,j\leq N}J_{i,j}x_{i}x_{j}$ of the model (Equation 1) only captures pairwise dependence between the binary variables, arising from an underlying network/interaction structure. Unfortunately, in most real-life scenarios, pairwise interactions are not enough to explain all of the complex dependencies arising in network data, and one has to take into account higher-order interactions arising from peer-group effects. Multi-body interactions are also common in many other branches of science; for example in Chemistry [6], it is known that the atoms on a crystal surface do not just interact in pairs, but in triangles, quadruples, and higher-order tuples. A natural extension of the classical 2-spin Ising model with a focus on capturing higher-order dependencies is the *p*-spin Ising model [7,8,9,10], where the quadratic interaction term in the sufficient statistic is replaced by a multilinear polynomial of degree p≥2. The probability mass function of this model is given by the following:(2)$$P_{\beta ,p}\left(x\right):=\frac{\exp\left\{-\beta H_{N}\left(x\right)\right\}}{Z_{N}\left(\beta ,p\right)}\;\mathrm{for}\;x\in {\left\{-1,1\right\}}^{N}$$
where the Hamiltonian $H_{N}\left(x\right)$ is given by the following:$$H_{N}\left(x\right):=-\sum_{1\leq i_{1},\ldots ,i_{p}\leq N}J_{i_{1}\ldots i_{p}}x_{i_{1}}\ldots x_{i_{p}}\;,$$
$J:={\left(\left(J_{i_{1}\ldots i_{p}}\right)\right)}_{\left(i_{1},\ldots ,i_{p}\right)\in {\left[N\right]}^{p}}$ is the interaction tensor (often taken as the scaled adjacency tensor of a hypergraph), β>0 denotes the interaction strength (or inverse temperature), *p* is the interaction order, and $Z_{N}\left(\beta ,p\right)$ is the normalizing constant (or partition function).

The *p*-spin Ising models have applications, including their role in explaining the microscopic theory of magnetism in solid and fluid ^3^He films absorbed on the surface of graphite (see [11]). They also serve as important resources for analog quantum simulation and quantum computation, contribute to the mapping of fermions to artificial spin systems in quantum algorithms for quantum chemistry, and appear in spin models of cuprate superconductor Hamiltonians (see [12] and the references therein). Furthermore, they play crucial roles in error suppression schemes for quantum annealers, adiabatic topological quantum computation, and other related areas. *p*-spin Ising and spin glass models have also appeared in a number of classical and recent works, such as [7,8,10,13,14,15,16,17,18] and *p*-spin versions of the closely related Potts model have appeared in [19,20,21].

The estimation of the parameter β in the model (Equation 2), assuming that the interaction order *p* is known, has been studied exptensively in the past (see, for example, [22,23,24] for the case p=2 and [9,10,16] for the case p≥3). For the p=2 case, $\sqrt{N}$-consistency of the so called *maximum pseudolikelihood estimator (MPLE)* of β under some general assumptions on the model (Equation 2) was first established in [23]. These assumptions can be verified easily for Ising models on many dense graphs at low temperatures (high values of β), and for Ising models on bounded-degree graphs as well as popular spin-glass models such as the Sherrington-Kirkpatrick model at all temperatures. Some of these results were extended in [22] where different rates of consistency were obtained for Ising models on sparse graphs, such as the sparse Erdös–Rényi graphs or sparse regular graphs, even at high temperatures. In addition, inestimability of β was established in [22] for Ising models on a sequence of dense graphs converging to a graphon for all β below a certain threshold, given by the inverse spectral norm of the limiting graphon. Precise central limit theorems for the *maximum likelihood estimator (MLE)* of β were derived for the 2-spin Ising model on complete graphs (also known as the Curie–Weiss model) in [25]. Some of these results have been extended in [9,10,16] for the case p≥3.

Now, suppose that one is given a sample X from a *p*-spin Ising model (Equation 2), with no other information on any of the parameters. A natural question is whether it is possible to estimate the interaction order *p* of the model. This is highly relevant from a practical perspective, as in many real-life applications such as recommender systems, strong peer-group effects exist, and two-interaction models are known to fit the data much worse than higher-order interaction models. For example, in [26], it was empirically shown that the 3-spin Ising model fitted the Last.fm music (accessed on 9 December 2020) (http://millionsongdataset.com/lastfm/) much better than the classical 2-spin model, thereby indicating the presence of complex peer-group effects in this dataset. This is a fan network database for a number of popular artists and bands, and the data consist of a list of binary opinions from the fan base for each artist, indicating whether each user is a fan of that artist or not as well as the user friendship network. The exact procedure involved first estimating the parameter β assuming a 2-spin Ising model from the data using the *maximum pseudolikelihood estimator*, followed by simulating a number of samples from the corresponding 2-spin Ising model with the estimated β as the parameter. It was observed that for most artists under consideration, the true Hamiltonian from the data lies outside the 2.5% and 97.5% quantiles of the histogram of the sampled Hamiltonians, thereby suggesting misfit. When a 3-spin Ising model was fit with the triangles in the user network as the hyperedges, the true Hamiltonian actually fell comfortably within the bulk of the histogram, thereby suggesting a good fit. In a different context, in [11], the authors mentioned that experimental and theoretical data indicated the relevance of three, four, five and six-spin exchange processes over a wide density range in adsorbed ^3^He films.

One might, in such situations, be interested in figuring out a systematic way of determining the value of the interaction order *p* for which the corresponding *p*-tuple interaction model fits the data best. To the best of our knowledge, this reverse problem of estimating the interaction order *p* given a single observation from the model (Equation 2) even with known β has not been addressed in the literature before. This question may be quite difficult for arbitrary underlying interaction tensors J, which necessitates some convenient structural assumptions on this tensor.

In this paper, we assume that tensor J has all entries equal to $N^{1-p}$, which corresponds to the *p*-spin Curie–Weiss model [7,9,27], given by the following:(3)$$P_{\beta ,p}\left(x\right):=\frac{\exp\left\{\beta N{\bar{x}}^{p}\right\}}{Z_{N}\left(\beta ,p\right)}\;\mathrm{for}\;x\in {\left\{-1,1\right\}}^{N}$$
where $\bar{x}:=\frac{1}{N}\sum_{i=1}^{N}x_{i}$. Even under this structural assumption, the possibility of estimating *p* depends on whether the interaction strength β is known or not. We will show that consistent estimation of *p* is impossible if β is unknown, which will be a consequence of our argument on the impossibility of the joint consistent estimation of β and *p*. At the heart of these impossibility results is the fact that the sufficient statistic $\bar{X}$ in the model (Equation 3) converges to a mixture of point masses at the maximizers of a certain function $H_{\beta ,p}$, and that these maximizers do not uniquely determine the tuple (β,p). This idea is formalized by a contiguity argument between the Curie–Weiss measures and *N*-fold products of Rademacher distributions, with the largest maximizer of $H_{\beta ,p}$ as the mean. It should be mentioned here that the related problem of joint inestimability of (β,h) for the classical 2-spin Curie–Weiss model with an additional magnetic field parameter *h* was addressed in [24] using similar contiguity arguments. We also extend these impossibility results to the more general *p*-spin Erdös–Rényi Ising models.

The situation is more intricate when β is known. In this case, we will show that there exists a strictly increasing threshold function $\beta^{*}\left(p\right)$, such that the consistent estimation of *p* is impossible for $\beta^{*}\left(p\right)>\beta$. However, for *almost all*
β (to be precise, for all but possibly countably many β), *p* can be estimated consistently whenever $\beta^{*}\left(p\right)<\beta$. The question of exactly describing the exceptional set of countably many βs for which the region $\beta^{*}\left(p\right)<\beta$ is inestimable for *p*, is still open. Finally, we want to mention that although joint consistent estimation of β and *p* is impossible using just one sample from the *p*-spin Curie–Weiss model (Equation 3), we still hope to do so using multiple samples.

## 2. Impossibility of Jointly Estimating (β,p)

We start by showing that joint consistent estimation of β and *p* using only one sample $X:=(X_{1},\ldots ,X_{N})$ from the model (Equation 3) is, in general, impossible. Towards this, for every m∈[0,1), define a set:$$\Theta_{m}:=\left\{\left(\beta ,p\right)\in \left(0,\infty \right)\times D:m\;\mathrm{is}\;\mathrm{the}\;\mathrm{unique}\;\mathrm{non}\text{-}\mathrm{negative}\;\mathrm{global}\;\mathrm{maximizer}\;\mathrm{of}\;H_{\beta ,p}\right\}$$
where *D* denotes the set of all integers ≥3, and$$H_{\beta ,p}\left(x\right):=\beta x^{p}-\frac{1}{2}\left\{(1+x)\log(1+x)+(1-x)\log(1-x)\right\}\;\mathrm{for}\;x\in \left[0,1\right]\;.$$

**Theorem** **1.**
*For every m∈[0,1), such that $|\Theta_{m}|\geq 2$, there does not exist any sequence of estimators (measurable functions of X), which is consistent for $\left(\beta ,p\right)\in \Theta_{m}$ under the model (Equation 3).*


The following lemma is crucial for proving Theorem 1.

**Lemma** **1.**
*For every m∈[0,1), denote the distribution of a Rademacher random variable with mean m by μ. Then, the product measure $Q:={⊗}_{i=1}^{N}\mu$ is contiguous to $P_{\beta ,p}$ for all m∈[0,1), and all $\left(\beta ,p\right)\in \Theta_{m}$.*


**Proof.** To begin with, note that on the event $E_{K}:=\left\{\sqrt{N}\left(\bar{X}-m\right)\leq K\right\}$, we have:$$\begin{array}{ccc} \frac{Q(x)}{P_{\beta ,p}\left(x\right)} & = & \frac{Z_{N}\left(\beta ,p\right)\prod_{i=1}^{N}{\left(\frac{1+m}{2}\right)}^{(1+X_{i})/2}{\left(\frac{1-m}{2}\right)}^{(1-X_{i})/2}}{\exp(\beta N{\bar{X}}^{p})}\\ & = & \frac{2^{-N}Z_{N}\left(\beta ,p\right)}{\exp\left(N\left[\beta {\bar{X}}^{p}-\frac{1}{2}\left\{\left(1+\bar{X}\right)\log\left(1+m\right)+\left(1-\bar{X}\right)\log\left(1-m\right)\right\}\right]\right)}\;. \end{array}$$
Now, by Lemma 3.2 and Lemma 3.4 in [9], we have:$$2^{-N}Z_{N}\left(\beta ,p\right)=e^{NH_{\beta ,p}\left(m\right)}\Theta \left(1\right)\;.$$
Also, by Taylor expansion, we have:$$\begin{array}{ccc} & & \beta {\bar{X}}^{p}-\frac{1}{2}\left\{\left(1+\bar{X}\right)\log\left(1+m\right)+\left(1-\bar{X}\right)\log\left(1-m\right)\right\}-H_{\beta ,p}\left(m\right)\\ & = & H_{\beta ,p}\left(\bar{X}\right)-H_{\beta ,p}\left(m\right)+\frac{1}{2}\left\{\left(1+\bar{X}\right)\log\left(\frac{1+\bar{X}}{1+m}\right)+\left(1-\bar{X}\right)\log\left(\frac{1-\bar{X}}{1-m}\right)\right\}\\ & = & O\left({\left(\bar{X}-m\right)}^{2}\right)+\left(\bar{X}-m\right)\left(\frac{1+\bar{X}}{1+m}-\frac{1-\bar{X}}{1-m}\right)=O\left({\left(\bar{X}-m\right)}^{2}\right). \end{array}$$
Hence, we have:$$\frac{Q(X)}{P_{\beta ,p}\left(X\right)}=\Theta \left(1\right)e^{NO\left({\left(\bar{X}-m\right)}^{2}\right)}\leq C_{K}$$
for some constant $C_{K}$. Hence, for any event $A_{N}\subseteq {\left\{-1,1\right\}}^{N}$, we have:$$Q\left(X\in A_{N},E_{K}\right)=E_{P_{\beta ,p}}\left[\frac{Q(X)}{P_{\beta ,p}\left(X\right)}1_{X\in A_{N},E_{K}}\right]\leq C_{K}P\left(X\in A_{N},E_{K}\right),$$
thereby providing the following:$$Q\left(X\in A_{N}\right)\leq C_{K}P\left(X\in A_{N}\right)+Q\left(E_{K}^{c}\right)\;.$$
Now, suppose that $P(X\in A_{N})\to 0$. Then, for every K>0, we have the following:$$\underset{N\to \infty }{lim sup}Q\left(X\in A_{N}\right)\leq \underset{N\to \infty }{lim sup}Q\left(\sqrt{N}\left(\bar{X}-m\right)>K\right)=P\left(N\left(0,1-m^{2}\right)>K\right).$$
Taking the limit as K→∞ throughout the above inequality, we can conclude that $Q(X\in A_{N})\to 0$, which completes the proof of Lemma 1. □

With Lemma 1 in hand, we are now ready to prove Theorem 1.

**Proof** **of Theorem 1**Suppose that there exists a sequence $T_{N}\left(X\right)\in R^{2}$ of consistent estimators of (β,p) on $\Theta_{m}$. Fixing two different points $\left(\beta_{1},p_{1}\right)$ and $\left(\beta_{2},p_{2}\right)\in \Theta_{m}$, we can construct disjoint neighborhoods $B_{1}$ and $B_{2}$ around them, respectively. Through the consistency of $T_{N}\left(X\right)$ on $\Theta_{m}$, we have the following:$$P_{\beta_{i},p_{i}}\left(T_{N}\left(X\right)\in B_{i}\right)\to 1\;\mathrm{for}\;i=1,2.$$
From Lemma 1, we have the following:$$Q(T_{N}\left(X\right)\in B_{i})\to 1\;\mathrm{for}\;i=1,2,$$
which contradicts the facts that $B_{1}$, $B_{2}$ are disjointed, and Q is a probability measure. □

**Remark** **1.**
*Our argument implies that the consistent estimation of p is impossible if β is unknown. Because, if there were such a consistent estimator $\hat{p}:=\hat{p}\left(X\right)$, then we could choose $\left(\beta_{1},p_{1}\right)$ and $\left(\beta_{2},p_{2}\right)$ from some $\Theta_{m}$; construct disjoint open intervals $I_{1}$ and $I_{2}$ around $p_{1}$ and $p_{2}$, respectively; and argue from contiguity that $Q(\hat{p}\left(X\right)\in I_{i})\to 1$ for i=1,2, which is a contradiction.*


The question is how do the sets $\Theta_{m}$ look for different values of m∈[0,1)? To answer this, let us define for each (β,p),$$\beta^{*}\left(p\right):=\sup\left\{\beta >0:\underset{x\in [0,1]}{\sup}H_{\beta ,p}\left(x\right)=0\right\}\;.$$
It follows from [9] that for p∈D, the function $H_{\beta ,p}$ has a unique postitive maximizer $m_{*}\left(\beta ,p\right)$ if $\beta >\beta^{*}\left(p\right)$. By convention, we define $m_{p}=m_{*}\left(\beta^{*}\left(p\right),p\right):=\lim_{\beta \to \beta^{*}{\left(p\right)}^{-}}m_{*}\left(\beta ,p\right)$.

**Proposition** **1.**
*The sequence ${\left\{m_{p}\right\}}_{p\in D}\to 1$ as p→∞, which can be sorted in ascending order as $0<m_{p_{1}}\leq m_{p_{2}}\leq \ldots <1$. Further, if we denote $\Theta_{m}{|}_{2}$ to be the projection of $\Theta_{m}$ onto the p-coordinate, then*

$$\Theta_{m}{|}_{2}=\left\{\begin{array}{cc} Ø & \;\mathrm{if}\;0<m\leq m_{p_{1}},\\ \left\{p_{1},\ldots ,p_{k}\right\} & \;\mathrm{if}\;m_{p_{k}}<m\leq m_{p_{k+1}},\;k\geq \;1. \end{array}\right.$$



The proof of Proposition 1 is technical, and is given in Appendix A. Note that for m∈(0,1), the set $\Theta_{m}$ is uniquely determined by its projection $\Theta_{m}{|}_{2}$, because every $p\in \Theta_{m}{|}_{2}$ uniquely corresponds to the element $\left(\beta_{p},p\right)\in \Theta_{m}$, where $\beta_{p}:=p^{-1}m^{1-p}\tanh^{-1}\left(m\right)$. Proposition 1 thus provides a complete description of the family of sets $\Theta_{m}$ for all values of m∈(0,1), and states that although each of these sets is finite, we can choose sets as large as possible from this family. The following proposition describes the set $\Theta_{0}$.

**Proposition** **2.**
*$\Theta_{0}=\left\{\left(\beta ,p\right)\in \left(0,\infty \right)\times D:\beta <\beta^{*}\left(p\right)\right\}$.*


The proof of Proposition 2 follows from the following three facts proved in [9]:If $\beta <\beta^{*}\left(p\right)$, then 0 is the unique global maximizer of $H_{\beta ,p}$.If $\beta =\beta^{*}\left(p\right)$, then $H_{\beta .p}$ has two different non-negative global maximizers.If $\beta >\beta^{*}\left(p\right)$, then $H_{\beta ,p}$ has a unique non-negative global maximizer, which happens to be positive.

**Remark** **2.**
*All of the results in this section also hold almost surely for the somewhat more general p-spin Erdös–Rényi Ising model [27], where the interaction tensor J in (Equation 2) is given by $\alpha^{-1}N^{1-p}A$, with the entries $A_{i_{1},\ldots ,i_{p}}$ of the tensor A being i.i.d. Bernoulli random variables with a mean α, for some fixed α∈(0,1). This follows from the fact that the Erdös–Rényi Ising and Curie–Weiss measures are mutually contiguous, which follows from Lemma 6.6 in [28].*


## 3. Estimation of p When β Is Known

Throughout this section, we assume that the parameter β is known. To begin with, for every β>0, let us define the following two sets:$$U_{\beta }:=\left\{p\geq 2:\beta^{*}\left(p\right)>\beta \right\}\;\mathrm{and}\;L_{\beta }:=\left\{p\geq 2:\beta^{*}\left(p\right)<\beta \right\}.$$
For Lemma A.1 in [10], the set $U_{\beta }$ is empty if β≥log2, and is of the form {q,q+1,…} for some integer q≥2 otherwise.

**Theorem** **2.**
*When β is known, no sequences of estimators that are consistent with $p\in U_{\beta }$ exist.*


**Proof.** It follows from Proposition 2 and Lemma 1, that the *N*-fold product measure of the mean-0 Rademacher distribution is contiguous to $P_{\beta ,p}$ for all $p\in U_{\beta }$. The proof now follows from the argument given in Remark 1. □

**Remark** **3.**
*One can consider an extension of the model (Equation 3) by adding an external magnetic field parameter h as follows:*

(4)
$$P_{\beta ,p}\left(x\right):=\frac{\exp\left\{\beta N{\bar{x}}^{p}+hN\bar{x}\right\}}{Z_{N}\left(\beta ,p\right)}\;\mathrm{for}\;x\in {\left\{-1,1\right\}}^{N}$$

*It has been shown in [9] that the consistent estimation of h is always possible when β is known. Proposition 2 shows that this is not the case for estimating p, which is impossible if $\beta^{*}\left(p\right)>\beta$. This inestimability region also coincides with that for estimating β when p is known (see [10]).*


Now, we turn our attention to the set $L_{\beta }$, and we break it down to the following two parts:$$L_{\beta }^{1}:=\left\{p\in L_{\beta }:\exists \;q\in L_{\beta },\;q\neq p\;\mathrm{such}\;\mathrm{that}\;m_{*}\left(\beta ,q\right)=m_{*}\left(\beta ,p\right)\right\}\;\mathrm{and}\;L_{\beta }^{2}:=L_{\beta }\setminus L_{\beta }^{1}\;.$$
where $m_{*}\left(\beta ,q\right)$ is defined as the largest non-negative maximizer of $H_{\beta ,q}$.

**Theorem** **3.**
*When β is known, there does not exist any sequence of estimators that is consistent for $p\in L_{\beta }^{1}$.*


**Proof.** Once again, from Lemma 1, we know that the *N*-fold product of the Rademacher distribution with a mean of $m_{*}\left(\beta ,p\right)$ is contiguous to both $P_{\beta ,p}$ and $P_{\beta ,q}$, where $p\in L_{\beta }^{1}$ and $p\neq q\in L_{\beta }$ is such that $m_{*}\left(\beta ,q\right)=m_{*}\left(\beta ,p\right)$. Clearly, $q\in L_{\beta }^{1}$. Once again, the rest of the proof follows from the arguments of Remark 1. □

We now show that it is possible to estimate *p* consistently on the set $L_{\beta }^{2}$, if β is known. Towards this, let us define the following estimator of *p*:(5)$$\hat{p}\left(\beta ,\delta \right):=\arg\min_{2\leq q\leq \log_{|\bar{X}{|}^{-1}}\left(2\beta +\delta \right)+2\;,\;\beta^{*}\left(q\right)<\beta }\left|m_{*}{\left(\beta ,q\right)}^{2}-{\bar{X}}^{2}\right|$$
The intuition behind constructing our estimator, is that ${\bar{X}}^{2}\overset{P}{\to }m_{*}{\left(\beta ,p\right)}^{2}$ as N→∞; hence, the distance between ${\bar{X}}^{2}$ and $m_{*}{\left(\beta ,q\right)}^{2}$ is expected to be minimized at q=p for large *N*. The next theorem shows that with a high probability, this is indeed the case.

**Theorem** **4.**
*For every δ>0 and $p\in L_{\beta }^{2}$, we have*

$$P_{\beta ,p}\left({\hat{p}}_{\beta ,\delta }=p\right)=1-e^{-CN}$$

*for some constant $C=C_{\beta ,p,\delta }>0$ not depending on N.*


**Proof.** Suppose that $X∼P_{\beta ,p}$ for some $p\in L_{\beta }^{2}$. It follows from Lemma A.1 in [10] and the proof of Proposition 1 that as q→∞, $m_{*}\left(\beta ,q\right)\to 1$ if β≥log2 and $m_{*}\left(\beta ,q\right)\to 0$ otherwise. Hence, in any case, $m_{*}\left(\beta ,p\right)\in \left(0,1\right)$ is not an accumulation point of the sequence ${\left\{m_{*}\left(\beta ,q\right)\right\}}_{q\geq 2}$. This shows that there exists ε>0, such that $|m_{*}{\left(\beta ,q\right)}^{2}-m_{*}{\left(\beta ,p\right)}^{2}|>\varepsilon$ for all $q\in L_{\beta }$ not equal to *p*.Now, it follows from the proofs of Lemma 3.1 and Lemma 3.3 in [9] that$$P_{\beta ,p}\left(|m_{*}{\left(\beta ,p\right)}^{2}-{\bar{X}}^{2}|>\frac{\varepsilon }{3}\right)\leq e^{-C_{1}N}$$
for some constant $C_{1}>0$ not depending on *N*. Also, it follows from the proofs of Lemma 3.1 and Lemma 3.3 in [9] and the proof of Proposition 1, that$$P_{\beta ,p}\left({\bar{X}}^{2}<{\left(2\beta +\delta \right)}^{2/(2-p)}\right)\leq e^{-C_{2}N}\;i.e.\;P_{\beta ,p}\left(p>\log_{|\bar{X}{|}^{-1}}\left(2\beta +\delta \right)+2\right)\leq e^{-C_{2}N}$$
for some constant $C_{2}>0$ not depending on *N*. Define $E_{1}:=\left\{\right|m_{*}{\left(\beta ,p\right)}^{2}-{\bar{X}}^{2}\left|\leq \varepsilon /3\right\}$ and $E_{2}:=\left\{p\leq \log_{|\bar{X}{|}^{-1}}\left(2\beta +\delta \right)+2\right\}$. It is clear that $\hat{p}\left(\beta ,\delta \right)=p$ on the event $E_{1}⋂E_{2}$. Theorem 4 now follows, as $P\left(E_{1}^{c}⋃E_{2}^{c}\right)\leq e^{-C_{1}N}+e^{-C_{2}N}$. □

**Remark** **4.**
*One can tune the parameter δ>0 in (Equation 5) to allow for an optimal number of integers over which the minimization is to be performed, in order to ensure proper convergence of $\hat{p}\left(\beta ,\delta \right)$. Also, in case β<log2, eventually $\beta^{*}\left(q\right)>\beta$, so one just stops at the largest q, for which $\beta^{*}\left(q\right)<\beta$ in the minimization (Equation 5).*


In the reverse problem of estimating β with known *p*, consistent estimation is possible for all $\beta >\beta^{*}\left(p\right)$ (see [9]). In contrast, a further challenging inestimability region $L_{\beta }^{1}$ arises in our problem of estimating *p* with known β. The next proposition shows that for almost all β>0, the set $L_{\beta }^{1}$ is actually empty, i.e., the entire region $L_{\beta }$ is estimable.

**Proposition** **3.**
*There exists a countable set Ξ⊂(0,∞), such that $L_{\beta }^{1}=Ø$ for β∉Ξ.*


The proof of Proposition 3 is given in Appendix B. However, in the proof we provide an exact enumeration of one countable set Λ satisfying Proposition 3, namely:$$\Lambda :=\left\{s_{p,q}:q>p\geq 2\;\mathrm{are}\;\mathrm{integers}\right\}\;\;\mathrm{where}\;s_{p,q}:=\frac{1}{p}{\left(\frac{p}{q}\right)}^{\frac{1-p}{q-p}}\tanh^{-1}\left({\left(\frac{p}{q}\right)}^{\frac{1}{q-p}}\right),$$
our analysis did not allow us to dig in any further. In particular, the following question is open: 

**Open Problem:** Give an exact enumeration of the minimal countable set $\Xi_{0}=\left\{\beta >0:L_{\beta }^{1}\neq Ø\right\}$ satisfying Proposition 3. 

Note that if $\beta =s_{p,q}$ for some integers q>p≥2, then $m:={\left(p/q\right)}^{1/(q-p)}$ is a stationary point of both $H_{\beta ,p}$ and $H_{\beta ,q}$. We can actually say that $\beta \in \Xi_{0}$ if this stationary point turns out to be a global maximizer of both these functions. However, checking the last condition will likely involve more intricate analysis, and is left as an open question. 

## Data Availability

No data were used in this work.

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
