# Peer review of "Interaction Order Estimation in Tensor Curie–Weiss Models"

_entropy, 2025, doi:10.3390/e27030245_

Round 1
Reviewer 1 Report
Comments and Suggestions for Authors
The author shows that simultaneous estimation of p and beta , given a single state of the model, is not possible on a p-spin Curie-Weiss Ising model and on a p-spin ErdÅ‘s-Rényi graph Ising model. The author also shows that if beta is known, it would be possible to estimate p for certain range of beta dependent on p .
The estimation of p given a single state is of considerable practical interest since it determines from available data the number of interacting Ising spins in the model that best describes the data. It is claimed that this problem has not been considered before.
I have the following correction to point out: On Eqs (1), (2), and (3), it appears {-1, 1}^n)and it should be {-1, 1}^N.
I consider that the research reported in the manuscript is interesting and technically correct, and therefore I recommend the manuscript to be accepted.
Author Response
Comment 1: I have the following correction to point out: On Eqs (1), (2), and (3), it appears {-1, 1}^n and it should be {-1, 1}^N.
Response 1: Thank you very much for pointing this out! I have now changed (marked in red) accordingly in equations (1) (pg-1), (2) (pg-2) and (3) (pg-3). The same typo was also there in equation (4) (pg-6), which I changed. Moreover, I have now carefully checked other places and found no such similar typo. Thanks once again!
Reviewer 2 Report
Comments and Suggestions for Authors
The comments are contained in the attached file.

Round 2
Reviewer 2 Report
Comments and Suggestions for Authors
The authors took my comments into account, and the objectives of the article bacame clearer.
When the Hamiltonians are written down, we can see that the exponent in the distribution functions (1) and (2) should be given with the opposite sign, or the difinitions of Hamiltonians should be changed.
Author Response
Comment: When the Hamiltonians are written down, we can see that the exponent in the distribution functions (1) and (2) should be given with the opposite sign, or the definitions of Hamiltonians should be changed.
Answer: Thank you very much for pointing this out. Yes, we agree; typically the Hamiltonian is written as the negative of the quadratic or tensor form that we originally had. The model also changes accordingly with the negative sign. We now incorporated these changes in the main paper.